# The Effect of Interventions on Preventing Musculoskeletal Injuries Related to Nurses Work: Systematic Review

**DOI:** 10.3390/jpm13020185

**Published:** 2023-01-20

**Authors:** Armando David Sousa, Cristina Lavareda Baixinho, Maria Helena Presado, Maria Adriana Henriques

**Affiliations:** 1Escola Superior de Enfermagem São José de Cluny, 9000-535 Funchal, Portugal; 2Nursing Research, Innovation and Development Centre of Lisbon, 1600-190 Lisbon, Portugal; 3Escola Superior de Enfermagem de Lisboa, 1600-096 Lisabon, Portugal; 4Center for Innovative Care and Health Technology (ciTechCare), Polytechnic of Leiria, 2411-901 Leiria, Portugal

**Keywords:** nurses, preventive program, musculoskeletal diseases, intervention studies, occupational injuries

## Abstract

Background: The 12-month prevalence of musculoskeletal disorders related to work (MDRW) in nurses rests between 71.8% to 84%, so it is urgent to develop preventive intervention programs with the purpose of avoiding negative physical, psychological, socioeconomic, and working aspects. There are several intervention programs aimed at preventing musculoskeletal disorders related to work for nurses, but few have successfully proven results. Despite the evidence pointing to the benefits of multidimensional intervention programs, it is essential to determine which interventions have positive effects on the prevention of this kind of disorder to create an effective intervention plan. Aim: This review intends to identify the different interventions adopted in the prevention of musculoskeletal disorders related to work in nurses and to compare the effectiveness of these interventions, providing the appropriate and scientific basis for building an intervention to prevent musculoskeletal disorders in nurses. Method: This Systematic Review was guided by the research question, “What are the effects of musculoskeletal disorders preventive interventions on nursing practice?” and carried out in different databases (MEDLINE, CINAHL, and Cochrane Central Register of Controlled Trials, SCOPUS, and Science Direct). Later, the results were submitted to the eligibility criteria, the appraisal quality of the papers, and the data synthesis was performed. Results: 13 articles were identified for analysis. The interventions implemented to control the risk were: training patient-handling devices; ergonomics education; involving the management chain; handling protocol/algorithms; acquiring ergonomics equipment; and no-manual lifting. Conclusions: The studies associated two or more interventions, the majority of which (11 studies) were training-handling devices and ergonomics education, therefore emerging as the most effective instruments in the prevention of MDRW. The studies did not associate interventions that cover all risk factors (individual, associated with the nature of the work, organizational, and psychological aspects). This systematic review can help with making recommendations for other studies that should associate organizational measures and prevention policies with physical exercise and other measures aimed at individual and psychosocial risk factors.

## 1. Introduction

Musculoskeletal disorders are a complex health problem transversal to all sectors of activity worldwide. The European institutions with responsibility for health and work have expressed their concern and provided guidelines for their control due to the risk of these injuries becoming pandemic, with repercussions on the economy of the different countries, including the increasing costs in the health systems [1].

The musculoskeletal disorders related to work (MDRW) are cumulative traumas resulting from the decompensation between the functional capacity of the muscle and its execution and frequency, which can lead to occupational diseases [2]. Usually, their origin results from the combination of several categories, which adds complexity to the causal identification, as well as its association with work. The consequences for professionals are numerous, of which the following stand out: physical and psychological suffering, loss of income, increased risk of chronicity, the economic costs inherent to the treatment, and the underlying burnout. This negative effect extends to the business level, in the present, with a reduction in productivity and an increase in absenteeism, and, in the long term, with the commitment of productive capacity and, consequently, an increase in costs [2,3].

The prevalence of MDRW in nurses at 12 months ranges from 71.85% to 84% [4,5,6,7], but only a small percentage (9.39%) is on sick leave due to MDRW [7]. Many of MDRW’s preventive intervention programs have been developed, such as: patient handling and mobilization programs [8], ergonomic intervention [9], psychosocial guidance on the work relationship [10], health promotion and prevention interventions [11], and exercise and physical therapy [12]. However, the authors observe that the evidence on programs with isolated interventions is limited [13] and we don’t know the effectiveness of their operational results.

A systematic review suggested that physical exercise at the workplace is considered an activity able to prevent occupational musculoskeletal disorders being able to enhance the physical capacity of workers. However, some studies showed contrasting results about the reduction of low back pain symptoms following only physical exercise at the workplace [14]. This is not a surprise since, considering the numerous and different variables in nurses’ workplaces and the role of the onset of this disorder, it is likely that its prevention needs a multidimensional approach that uses the simultaneous adoption of technical, organizational, procedural, and training measures.

Another systematic review identified, based on their network meta-analyses, that low back exercises plus health education were the most effective procedures on the effects of non-drug intervention management in nurses, followed by single low back exercise intervention and yoga [15].

In conclusion, several reviews have been conducted by healthcare professionals and some of them assessed the effectiveness of interventions in nurses’ homes [16] and nurses [3,17]. Others weighed the risk of handling overweight and obese patients by nursing assistants [18] or focused on interventions for reducing low back pain in nurses [3], but we didn’t find reviews on the effect of interventions in nurses-midwives.

The design of a preventive intervention program for MDRW must be based on multidimensionality and the diagnosis of the specific needs of the target population, such as work characteristics, the type of ergonomic equipment existing, the environment, and the organizational culture implemented [19]. These types of preventive programs can be more successful due to their direct relationship with the praxis and should include the assessment of risk perception, educational programs with ergonomic posture training of preventive clinical skills, physical activity at the workplace, cognitive-behavioral therapy for the treatment of physical, psychological, occupational, ergonomic risk factors, and the promotion of a safe environment [19].

Changing an individual’s behavior or reducing task-specific risks has been the focus of most interventions, but rather the broader contextual factors that are associated with the complex ethology of MDRW, such as risk, adhesion to the hierarchical chain in risk control, lack of commitment from management, culture and organizational conditions, understanding the importance of worker participation, regulated legislative practice, and competence in risk management [20].

The literature review shows that the studies focus essentially on the epidemiology of injuries and those that explore the interventions adopt one or two interventions, not systematizing all the effective interventions to control this occupational health problem.

In view of the above, the aim of this review was to identify the different interventions adopted in the prevention of musculoskeletal disorders related to work in nurses and to compare the effectiveness of these interventions, providing the appropriate and scientific basis for building an intervention to prevent musculoskeletal disorders in nurses praxis.

## 2. Materials and Methods

### 2.1. Study Design

Given the purpose of the study and the state of the art of the phenomenon under study, a systematic review (SR) was chosen [21,22]. The option for a review with a scientific and systematic methodology is justified by the need to have reliable results from which conclusions can be drawn and decisions made, minimizing the risk of bias and guiding clinics and health policies based on research results [21,23]. The protocol to guide the SR [22] was prepared and agreed upon in December 2021, and it was registered on Prospero with ID No. CRD42022331581 in May 2022.

The research question that guided the definition of eligibility criteria and the research strategy was: What are the effects of MDRW preventive interventions on nursing practice?

### 2.2. Eligibility Criteria

Inclusion criteria were primary experimental and epidemiological study designs (RCT, non-RCT, quasi-experimental, cohort studies, case-control studies, and analytical cross-sectional studies), which measure interventions for the prevention and/or reduction of musculoskeletal disorders related to nurses’ work. There were no restrictions on country or year of research, but they were limited to studies published in Portuguese, English, and Spanish. Reports, such as unpublished manuscripts and conference abstracts, are not eligible for inclusion.

Exclusion criteria were systematic review, studies with qualitative design or protocols, studies in which the target population is only nursing assistants or home nurses, studies in which the intervention is for treatment or rehabilitation of injuries or illnesses of MDRW, and studies with intervention in the multidisciplinary team that does not identify nurses’ results.

The main outcomes appraised were incidence and prevalence of musculoskeletal disorders, absenteeism rate, related pain, back, upper limbs, shoulders, and neck loading, adherence to safe behaviors from a biomechanical point of view, and acceptance and adhesion to the program by nurses.

### 2.3. Research Strategy

The first stage of the search began in August 2021, carried out with natural language terms in association with the medical subject headings—MeSH and SCoR guidelines, conducted on the EBSCO host platform in the MEDLINE, and CINAHL databases. The terms were associated with the Boolean operators OR and AND at the junction of the descriptors identified in PICO.

This first research allowed us to identify the keywords and descriptors used by the indexing of articles and to raise awareness of current scientific knowledge, such as helping the elaboration of the study protocol.

The second stage of the research started in January 2022 and lasted until May of the same year; the survey was conducted on the EBSCO host platform (MEDLINE, CINAHL, and Cochrane Central Register of Controlled Trials), SCOPUS, and Science Direct.

The last research was conducted in May of 2022. An inspection of the bibliographic references of the articles was carried out to identify systematic reviews that report and guide the future research of systematic reviews for the important conclusions about the topic.

The strategy used in Medline Complete research is presented in Table 1.

### 2.4. Data Extraction, Quality Appraisal, and Data Synthesis

After identifying all the articles in the different databases, they were transferred to EndNote, Clarivate Analytics, Philadelphia, United States, to recognize duplicate articles and eliminate them. For the calculation of the relevance of the article, we transfer all articles in the “RIS” format to “Rayyan”.

The process of data extraction started by analyzing the title and synopsis of all articles based on the selection criteria initially defined. This process was carried out by the two reviewers independently, in case of doubts, and they were clarified through a third reviewer.

An excel file was built to extract the results, which was carried out by the same researchers. Each article was summarized and organized according to the following items: study identification (author, year of publication, and country), objective, type of study, sample, and results.

The risk of being biased, the instruments RoB2, quality assessment of before-after studies with no control group, and the quality assessment of papers describing observational and quasi-experimental studies that were used [21,24] were assessed by the team of researchers. Given the heterogeneous nature of the study designs, a narrative synthesis was chosen to answer the research question. For the quality of the evidence, we considered a confidence interval of 95%.

## 3. Results

A total of 48 duplicate articles (of the 165 articles submitted) were identified. 

After analyzing the title and synopsis, we excluded a total of 103 records for not complying with the inclusion criteria and were left with 13 articles to identify their eligibility.

The reasons for the exclusions of the 103 articles were: 28 articles were excluded for analyzing the wrong population (nursing assistants, home nurses, bus drivers, patients, orthodontists, industrial workers), 43 articles for wrong outcomes (treatment of chronic musculoskeletal disorders, prevalence, effects of the workplace, others occupational disorders: stress, anxiety, and dermatology), 31 articles for study design (systematic reviews, literature review, and protocols), and one for foreign language (Arabic writing).

Finally, we had 13 articles for analysis. The research diagram and the study selection process can be seen in Figure 1. 

One article was eliminated after the eligibility assessment for low quality of evidence. Of the selected articles, six are from the USA and three from Canada, the remaining articles are from China, Iran, Germany, and Vietnam.

Regarding the year of publication, we can see that the oldest article was published in 2001 and that 38.5% of the articles were published in the last five years (one article in 2017, one article in 2020, two in 2021, and one in 2022).

This diffusion of countries and the growing number of publications on this topic demonstrates the interest and importance that this topic has for the scientific community worldwide.

Table 2 presents the extraction of the results of the 13 articles, identifying the objectives, the type of study, the time of evaluation of the intervention, the evaluation instrument, and its results, and finally, the conclusions of the articles.

### 3.1. Quality of the Evidence

Three instruments were used to assess the bias of the 13 studies. Two RCTs have been evaluated by RoB2.0 (Table 3) and five studies for the quality assessment of before–after studies with no control group (Table 4) and the quality assessment of papers describing observational and quasi-experimental studies were the instrument use for the other six studies (Table 5).

The study by Pourhaji et al. [30], presented after its judgment through the ROB2: low risk of bias, and the study Yassi et al. [25], presents some concerns, since it presented in D3 some concerns.

### 3.2. Interventions to Prevent Musculoskeletal Disorders Related to Work in Nursing 

The studies included in this SR allow us to answer the research question. As can be seen in Table 6, the researchers associated two or more interventions [25,26,27,28,29,30,31,32,33,34,35,36], the most frequent being the training-handling devices, with the aim of training nurses to use the equipment for mobilization and/or patient transfer [19,25,26,27,28,29,31,32,33,34,35,36], thus controlling the major risk factor for injuries. 

Eleven studies associate training-handling devices with ergonomics education [19,25,26,27,28,29,31,32,33,36] and information on risk factors, use of mechanical devices, and use of the principles of body mechanics during the use of mechanical means, but also in carrying out other activities.

## 4. Discussion

The 12 studies in this review are heterogeneous from the point of view of study design, sample size, implemented intervention, assessment instruments, measured outcomes, and contexts where the study was carried out, which does not allow for meta-analysis. Nevertheless, the results allow the evaluation of the methodological quality of the studies and the evidence of the interventions that each study used to control the risk of musculoskeletal injury.

This SR made it possible to identify which interventions prevent MDRW in nurses and synthesize the evidence on which interventions were implemented, their feasibility, and their impact on different outcomes, with special relevance to the prevalence of MDRW. This type of injury is a global and transversal problem in almost all professions, but it assumes a worrying incidence and prevalence among health professionals, especially in nurses [23,24,25,26,27,28,29,30,31,32,33,34,35,36,37,38] due to the very nature of the professional activity, with the need to mobilize and transfer patients with a high degree of dependence, performing activities in positions that imply dorsiflexion and torsion of the spine, lifting weights above the recommended for the anthropometric characteristics of the professional [37,38,39,40], few rest periods between activities that demand of high physical effort, or the maintenance of painful postures for a long time [37,38,39].

The studies included reinforce the importance of programs aimed at learning the correct handling of patients and/or using the principles of biomechanics in carrying out this activity [19,25,26,27,28,29,30,31,32,33,36], the use of mechanical means that reduce overload, and the associated risk [19,25,26,27,28,29,31,32,33,34,35,36], including the policy of no manual lifting [27]. These data are consistent with the recommendations of the Occupational Safety and Health Administration (OSHA) which recommends the increased availability of assistive devices and the use of even the most basic assistive devices as an integral part of safe patient handling [40].

Only two programs involved the management chain [27,32], and the results show an impact of the intervention in reducing the incidence of injuries and absenteeism, which reinforces the authors’ recommendations for the development of a safety culture in institutions providing care for health, ensuring knowledge, skills, and competencies for the prevention of injuries in its professionals [36,38,40]. Although organizational cultural changes take time, this is equally applicable to safety culture [40], and it’s urgent that healthcare provider settings are increasingly dynamic work environments that benefit from strong organizational programs, policies, and practices around risk identification and reduction [39].

It should be noted that the results of some programs, in relation to the impact on the reduction of injuries and the adoption of safe behaviors, do not observe gains, for example, in the use of equipment or changes in practices [25,28], leaving the question of whether the involvement of the organization with clear policies for the clinic of professionals, guaranteeing the safety of the patient and the professionals, and adhesion to the programs would increase since the beneficial effects of safe patient-handling programs improved over time, which highlights the importance of a long-term and continued effort to make the necessary cultural changes [41].

We corroborate that interventions should take into account not only the ergonomics but also the improvement of the organizational aspects of the work environment [37], but other aspects should be explored, such as communication with the patient to actively involve them in the procedures and promote their rehabilitation, or even involving the caregiver [42], previously planning the activity by unblocking the space around the patient’s bed, ensuring an optimization of the interaction between the health professional and the patient [37] or his caregiver [40], and the health professional and the environment [36].

The studies evaluated two or more interventions simultaneously, which goes against the key idea that risk control is carried out through the implementation of systemic and multifactorial programs [37]. Multiple approaches are needed to put changes in practice and to promote a safety culture, including workflow processes, ongoing training, skills, supervising, and communication between professionals about risk [37,38,39,40]. It is necessary to encourage in the units the choice of facilitators to teach, change behaviors, and monitor the appropriate use of mobility aids [40] and also to promote adequate training to improve the knowledge and skills of the nursing staff in the handling of dependent patients [43]. 

Outcomes focused on incidence, prevalence, self-perceived frequency and intensity of physical discomfort, musculoskeletal pain, time-loss rates, risk perception, and perception of a safe working environment [19,25,26,27,28,29,30,31,32,33,34,35,36]. It is suggested that future research explores nurses’ adhesion to the programs. A study that aimed to provide a systematic review of the international literature, synthesize knowledge, and explore factors that influence nurses’ adhesion to patient-safety principles concluded that patients’ participation, healthcare providers’ knowledge and attitudes, a collaboration by nurses, appropriate equipment and electronic systems, education and regular feedback, and standardization of the care process influenced nurses’ adherence to patient-safety principles [44]. 

Interventions essentially focused on preventive measures aimed at risk and not at promoting the health of professionals. Given the multifactorial nature of risk factors for falls, individual, psychosocial, organizational, and socio-economic, we share the opinion of other studies that recommend preventive measures including physical exercise for muscle strengthening, food education to maintain weight, cognitive-behavioral strategies to control anxiety, and investing in a good work environment to control psychosocial risk factors [41,45,46,47,48,49,50].

Some studies identify the ineffectiveness of training only one factor [22], and others showed that multifactor training (transfer, lifting, and repositioning), and the multiple interventions (education and training, zero lift policy, provision of assistive devices for patient support and care, individual measure, etc.) are emerging as the most effective instruments in the prevention of MDRW [16,24,25,27,30].

These multidimensional intervention programs reduce the self-reported performance of “unsafe” working environments [16,24], decrease time-loss/injury days, modify duty days, increase job satisfaction, and decrease workers compensation costs [24,25,30,33]. A peer leader program is much more effective than traditional educational approaches and facilitates the implementation of the program, as well as being sustainable over time [24], especially in small hospitals [25]. 

The procede-proceed model has a significant effect on behaviors as a factor that increases the quality of lifestyles of low back pain (LBP) [27]. Theoretical education was effective in improving knowledge, attitude, and self-efficacy, reinforcing and enabling factors, and behavior immediately after 6–12 months of intervention [27]. Other studies conclude that the effective goal of reducing MDRW is the combining of theoretical education with ergonomics practice [33]. The social media approach to maintaining behavior for a long period of time (6 months) was more successful than the face-to-face approach [27].

### Study Limitations

The heterogeneous design of the quantitative studies, the different instruments used to evaluate the intervention, and their differences did not allow for meta-analysis and limits the evidence of this SR. In addition to this factor, the search was conducted only in four databases and the inclusion of studies in Portuguese, Spanish, and English may have excluded studies published in other languages that would have answered the research question.

## 5. Conclusions

The interventions implemented to control the risk of MDRW were training patient-handling devices, ergonomics education, involving the management chain, handling protocol/algorithms, acquiring ergonomics equipment, and no-manual lifting. The use of two or more interventions in association allowed a reduction in the prevalence of MDRW and associated symptomatology, increasing risk perception, decreasing frequency and intensity of physical discomfort, musculoskeletal pain, time-loss rates, and risk perception.

Combining theoretical education with ergonomics will be more effective in the goal of reducing MDRW, and the organization must implement appropriate policies to apply the intervention more effectively.

Future studies should associate organizational measures and prevention policies with physical exercise and other measures aimed at individual and psychosocial risk factors because the multifactorial nature of risk can only be controlled with multifactorial interventions with a combination of individual, psycho-organizational, and task-related measures.

In nursing education, both graduate and postgraduate, it´s important to introduce curricula content on risk factors and preventive measures, enabling the student to adopt these measures in clinical practice. The simulated practice in the laboratory can be a good pedagogical strategy for the development of these competencies.

For the professionals who are in the clinic, the simulation and video recording of the posture and movements performed in the provision of care can allow the awareness of the individual risk associated with the nature of the professional activity.

The methodological quality of the studies is acceptable and makes it possible to make recommendations for the clinic, for training, and for research. It should be noted that the heterogeneity of the program’s conditions, the robustness of the evidence, and the synthetic description of the interventions make it difficult to understand the whole of the program and, above all, how its implementation was carried out in terms of time, involvement of human resources, strategies for adherence to the program, and collaborative work within the contexts (or lack thereof).

## Figures and Tables

**Figure 1 jpm-13-00185-f001:**
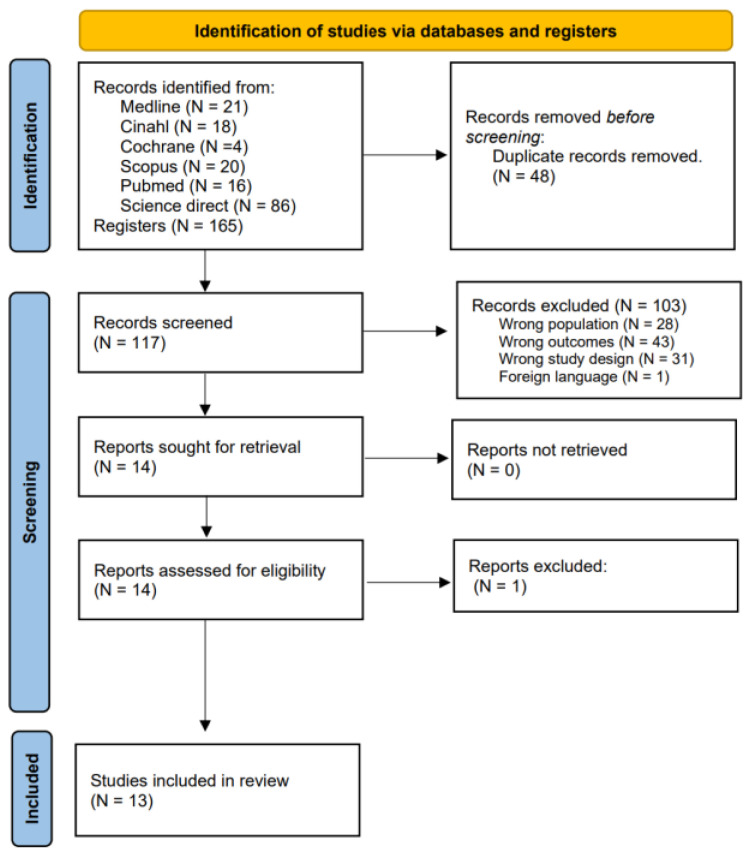
PRISMA Flow diagram.

**Table 1 jpm-13-00185-t001:** Search Strategy. Lisbon, 2022.

	Search Strategy	Number of Articles
**#1**	((((((((nursing[Title/Abstract]) OR (midwive[Title/Abstract])) OR (nurse-midwives[Title/Abstract])) OR (nurs*[Title/Abstract])) OR (midw*[Title/Abstract])) OR (nursing care[MeSH Terms])) OR (midwife[MeSH Terms])) OR (nurse midwife[MeSH Terms])) OR (nurse midwives[MeSH Terms]),,,”““nursing”“[Title/Abstract] OR ““midwive”“[Title/Abstract] OR ““nurse-midwives”“[Title/Abstract] OR ““nurs*”“[Title/Abstract] OR ““midw*”“[Title/Abstract] OR ““nursing care”“[MeSH Terms] OR ““midwifery”“[MeSH Terms] OR ““nurse-midwives”“[MeSH Terms] OR ““nurse-midwives”“[MeSH Terms]”	1,106,849
**#2**	(((((((((((((((((((((occupational injuries[Title/Abstract])) OR (injuries[Title/Abstract])) OR (occupational diseases[Title/Abstract])) OR (musculoskeletal injuries[Title/Abstract])) OR (musculoskeletal disorders[Title/Abstract])) OR (musculoskeletal disorders related to work[Title/Abstract])) OR (musculoskeletal pain[Title/Abstract])) OR (back pain[Title/Abstract])) OR (neck pain[Title/Abstract])) OR (tendinitis[Title/Abstract])) OR (back hernias[Title/Abstract])) OR (hernias of the spinal column[Title/Abstract])) OR (Inj*[Title/Abstract])) OR (Musculos* inj*[Title/Abstract])) OR (abnormalities, musculoskeletal[MeSH Terms])) OR (musculoskeletal diseases[MeSH Terms])) OR (musculoskeletal disease[MeSH Terms])) OR (back pain[MeSH Terms])) OR (tendinitis[MeSH Terms])) ) OR (back pain, low[MeSH Terms])”,,,”““occupational injuries”“[Title/Abstract] OR ““injuries”“[Title/Abstract] OR ““occupational diseases”“[Title/Abstract] OR ““musculoskeletal injuries”“[Title/Abstract] OR ““musculoskeletal disorders”“[Title/Abstract] OR ((““musculoskeletal diseases”“[MeSH Terms] OR (““musculoskeletal”“[All Fields] AND ““diseases”“[All Fields]) OR ““musculoskeletal diseases”“[All Fields] OR (““musculoskeletal”“[All Fields] AND ““disorders”“[All Fields]) OR ““musculoskeletal disorders”“[All Fields]) AND ““related to work”“[Title/Abstract]) OR ““musculoskeletal pain”“[Title/Abstract] OR ““back pain”“[Title/Abstract] OR ““neck pain”“[Title/Abstract] OR ““tendinitis”“[Title/Abstract] OR ((““back”“[MeSH Terms] OR ““back”“[All Fields]) AND ““hernias”“[Title/Abstract]) OR ((““hernia”“[MeSH Terms] OR ““hernia”“[All Fields] OR ““hernias”“[All Fields] OR ““hernia s”“[All Fields] OR ““herniae”“[All Fields]) AND ““the spinal column”“[Title/Abstract]) OR ““inj”“[Title/Abstract] OR (““musculos*”“[All Fields] AND ““inj”“[Title/Abstract]) OR ““musculoskeletal abnormalities”“[MeSH Terms] OR ““musculoskeletal diseases”“[MeSH Terms] OR ““musculoskeletal diseases”“[MeSH Terms] OR ““back pain”“[MeSH Terms] OR ““tendinopathy”“[MeSH Terms] OR ““low back pain”“[MeSH Terms]”,	4134
**#3**	((((((((((((((intervention[Title/Abstract]) OR (programme[Title/Abstract])) OR (program*[Title/Abstract])) OR (Int*[Title/Abstract])) OR (action[Title/Abstract])) OR (multidimensional intervention[Title/Abstract])) OR (organizational program[Title/Abstract])) OR (preventative measures[Title/Abstract])) OR (preventive actions[Title/Abstract])) OR (prevention measures[Title/Abstract])) OR (prev*[Title/Abstract])) OR (accident prevention[MeSH Terms])) OR (early intervention[MeSH Terms])) OR (measures[MeSH Terms])) OR (assistance program, employee health care[MeSH Terms])”,,,”““intervention”“[Title/Abstract] OR ““programme”“[Title/Abstract] OR ““program*”“[Title/Abstract] OR ““int”“[Title/Abstract] OR ““action”“[Title/Abstract] OR ““multidimensional intervention”“[Title/Abstract] OR ““organizational program”“[Title/Abstract] OR ““preventative measures”“[Title/Abstract] OR ““preventive actions”“[Title/Abstract] OR ““prevention measures”“[Title/Abstract] OR ““prev*”“[Title/Abstract] OR ““accident prevention”“[MeSH Terms] OR ““early intervention, educational”“[MeSH Terms] OR ““weights and measures”“[MeSH Terms] OR ““occupational health services”“[MeSH Terms]”	1,294,797
**#1 AND #2 AND #3**		21

**Table 2 jpm-13-00185-t002:** Type of outcomes and results of studies. Lisbon, 2022.

Study/ Year/Country	Aim	Type of Study/ Sample	Time of Evaluation of Intervention and Participants	Outcomes and Instruments	Conclusions
Yassi et al. (2001) Canada [25]	“to compare the effectiveness of training and equipment to reduce musculoskeletal injuries, increase comfort, and reduce physical demands on staff performing patient lifts and transfers at a large acute care hospital.”	Randomized controlled trial (RCT) (three-armed)	The baseline was on 1 July 1998, and 6-months, and again at 12-months. The participants 346 nurses	Outcome: The frequency of manual patient-handling tasks, i.e., those tasks during which neither mechanical nor nonmechanical assistive equipment is used, was significantly and markedly decreased in the Arm C “no strenuous lifting” wards by 6 months, by an average of 9 tasks per shift. This decrease was sustained at 1 year. In contrast, there was no significant change over 6 months in frequency of manual handling tasks on Arm A or Arm B. By the time of the 6-month follow-up, there was a significant increase in use of assistive devices such as transfer belts and sliding devices on Arm B; however, this increase was not sustained at 1 year. The use of these manual assistive devices increased significantly (P 5 0.021) by 6 months on Arm C, but by 1 year use declined significantly. At 6 months nurses on Arm C reported using the sit-stand lift an average of 4.9 times per shift. This declined significantly to 3.2 times per shift at 1 year. The use of total body lifting equipment over time varied significantly by service type. Instruments: Visual analogic scales (VAS); SF36; Oswestry Low Back Pain Disability Questionnaire; Disability of Arms, Shoulder, and Hands (DASH).	After 6 months increased the using of lifting devices, but there was a decrease in one year. The “no strenuous lifting” program, which combined training with assured availability of mechanical and other assistive patient handling equipment, most effectively improved comfort with patient handling, decreased staff fatigue, and decreased physical demands. The fact that injury rates were not statistically significantly reduced may reflect the less sensitive nature of this indicator compared with the subjective indicators.
Owen et al. (2002) USA [26]	“to determine the impact of an Ergonomic program on perceived stress ratings, injury rates and patient care.”	Quasi experimental study design	18-months pre- intervention, 18-months after intervention, and five years follow up period Participants: 319 data collection by nursing staff.	Outcomes: “There were 319 data collection forms completed by the nursing staff after carrying out the patient handling tasks. More than half (*n* = 182) involved transferring patients from bed to chair/commode and back to bed. (…) The mean of perceived exertion ranged from 0.6 to 1.0 for the shoulder by experimental site subjects and from 3.2 to 5.2 for control site nursing personnel. (…) The mean of perceived exertion to the lower back ranged from 0.5 to 0.7 by experimental site subjects and from 3.3 to 5.0 for control site nursing personnel.” Instruments: Two Likert-type scale, the OSHA 200 forms and the incident report forms generic to the hospitals	The perceived physical exertion was significantly reduced for all tasks at the experimental hospital. The number of back injuries, lost work, and restricted days have also decreased. If one compared the data of 18 months pre intervention to the data of 18 months post intervention, the injuries decreased to 40%, the LWDs decreased from 64 down to 3, and restricted days decreased to 20%. The patients felt more comfortable and more secure when assistive devices were used.
Nelson et al. (2006) USA [27]	(1) to design and implement a multifaceted program that successfully integrated evidence-based practice, technology, and safety	Pre-/post design without a control group (Focus groups)	Nine-month pre- intervention (May 2001–January 2002) and the nine-month post-	Outcomes: Key measures included injury rate, lost and modified workdays, job satisfaction, self-reported unsafe patient handling acts, level of support for program, staff and patient acceptance, program effectiveness, costs, and return on investment.	This multi-faceted program resulted in positive outcomes associated with injury rates, modified duty days, job satisfaction, self-reported safety in performing patient handling
	improvement; (2) to evaluate the impact of the program on injury rate, lost and modified workdays, job satisfaction, self-reported unsafe patient handling acts, level of support for program, staff and patient acceptance, program effectiveness, costs, and return on investment.		intervention (February 2002–October 2002) Participants: 825 nurses	Post-intervention injury rate decreased in 15 of the 23 units, increased in 7 units and remained the same in 1 unit. Overall, the injury rate decreased from 24.0/100 caregivers at baseline and 16.9/100 caregivers’ post-intervention. Post-intervention injury rates were found to be significantly lower X^2^ (1, *n* = 46) = 4.42, *p* = 0.036. The number of modified duty days decreased significantly (*p* = 0.02) from 1777 modified duty days during the 9-month pre-intervention period, to 539 modified duty days during the 9-month post-intervention period. Instruments: Poisson regression model to test differences pre- and postintervention, Mean values for the number of modified duty days and lost days taken per injury, and survey results were interpreted using the modified Bonferoni approach.	tasks, and cost. The program was well accepted by patients, nursing staff, and administrators. While the total number of lost workdays decreased by 18% post-intervention, this difference was not statistically significant
Black et al. (2011) Canada [28]	To evaluate the effectiveness of a Transfer, Lifting and Repositioning (TLR) program to reduce musculoskeletal injuries (MSI) among direct health care workers,	Retrospective pre and post-intervention design, utilizing a nonrandomized control group	One year pre- and post-intervention. (September 2002–June 2004; January to December 2005) Participants: 766 TLR injuries cases	Outcomes: The number of injuries by occupation showed that in the control group, the distribution remained unchanged with the exception of therapists (physical therapists, occupational therapists, respiratory therapists) where a significant increase was seen. The most significant change was seen in the decrease in injuries in attendants (from 25.4% to 0%) and increase in injuries in nurse aides (from 1.1% to 11.3%) in the intervention group. Instruments: Analysis of all injuries and time-loss rates.	Significantly reduce both time-loss and no-time-loss injuries and disability related to patient handling. The reductions of claim costs/injury represented a substantial benefit to the intervention hospitals. The program intervention seemed to be more effective in the small hospitals than in the medium or large ones.
Yang et al. (2021) China [19]	To evaluate the effectiveness of a multidimensional intervention program to prevent and reduce WRMDs in ICU nurses.	Quasi-experimental cluster-randomized controlled trial	Baseline, 3 and 6 months after the start of the intervention (December 2017, to January 2018) Participants: 201 nurses	Outcomes: A total of 201 nurses from four mixed ICUs in four hospitals were recruited. From two ICUs, 94 nurses were assigned to the intervention group, and from the two remaining ICUs, 104 nurses were assigned to the control group by cluster random sampling. Ultimately, 190 nurses provided three recorded outcome measurements (intervention group, *n* = 89 [94.68%]; control group, *n* = 101 [94.39%]). No significant difference in loss to follow-up was found between the two groups (χ^2^ = 0.074, *p* = 0.862). GEE showed that the multidimensional intervention program improved the risk perception of WRMDs (OR = 0.517, *p* < 0.001) and health behavior application (OR = 0.025, *p* < 0.001), relative to that of the routine specialist training. Interactions between the measurement time and group were observed (*p* < 0.001). Age and the length of ICU employment affected the perception of a safe working environment (*p* = 0.047 and *p* = 0.011 respectively). The GEE, including age and ICU employment, indicated that the measures of the intervention group and the control group were statistically significant. The perception of an unsafe working environment in the control group was 1.637 times that in the intervention group (OR = 1.637, *p* = 0.024).	A meticulous planning is essential to make interventions compatible with the daily work routine. The multidimensional intervention program seems applicable from time, financial, and organizational perspectives, he helped to reduce the short-term reported incidence rate of WRMDs, improve the nursing risk perception and health behavior application, and promote a safe working environment.
				Instruments: Baseline demographic information was collected, true self-reports on-line questionaries—Nordic Musculoskeletal Questionnaire (measure self-perceived symptoms of WRMDs), Chinese version of the Risk Perception of Musculoskeletal Injury (risk perception); Nursing Physical Factors Evaluation Questionnaire (Application of Health behavior); Hospital Safety Climate Questionnaire (Perception of a safe working environment).	
Zadvinskis et al. (2009) USA [29]	To examine the effectiveness of a multifaceted minimal-lift environment on reported equipment use, musculoskeletal injury rates, and workers’ compensation costs for patient-handling injuries.	A mixed measures design with both descriptive and quasi-experimental design	Baseline and 3 and 12 months after intervention (April 2007) Participants: 77	Outcomes: 46 were in the intervention group (53% survey response rate) and 29 (39% survey response rate) were in the control group. There were no significant demographic differences between participants in the intervention and control groups. The combined sample (*n* = 75) was 95% female and ranged in age from 21 to 59 years (mean 33.7). Nursing staff members from the multifaceted minimal-lift environment experienced a reduction in patient-handling injuries and costs compared to nursing staff working in a non–minimal-lift environment. The intervention unit injury incidence rate was 3.26/100 full-time equivalents (FTEs) whereas the control unit injury incidence rate was 3.43/100 FTEs. Injury costs for the intervention unit were $6566 compared with $11,145 for the control unit (a $4579 difference). After subtracting the cost of peer coach education ($1680), the intervention unit experienced a $2899 return on investment. Instruments: Demographic and equipment use data for the intervention and control units were collected through self-report via pen-and-paper survey.	Implementing a successful multifaceted minimal-lift environment for nursing staff can be time consuming and complex. Intervention strategies must match the innovation, target group, and workplace context, and could expand program elements to include ergonomic assessment protocols, after action reviews.
Pourhaji et al. (2020) Iran [30]	To investigate the effects of an educational program based on Precede-Proceed model on promoting Low Back Pain (LBP) behaviors among health care workers (HCWs).	A randomized trial	6 and 12-months follow-ups. Participants: 102	Outcomes: The present study was conducted on HCWs aged 30 to 55 (The Subjects 75 Intervention Into two group1, 2 and 37 control groups) in the comprehensive Service centers. The mean age of the intervention group was 46.34 ± 1.18, and the mean age of the control group was 47.23 ± 1.15 years (*p* = 0.598). Prior to the intervention, there was no significant difference between the two intervention groups and one control group. The repeated measure analysis test confirmed that was important and significant difference 6 and 12 months after the Intervention. There was a significant interaction between the factors “group” and “test time” (*p* < 0.05, *p* < 0.001). Increasing the mean score of attitude, knowledge, perceived self-efficacy, enabling factors, reinforcing factors, quality of life, public health, and preventive behaviors of LBP in intervention group (*p* < 0.05, < 0.001), but no significant change in mean score of knowledge, attitude, Self-efficacy, quality of life, general health, reinforcing	Health behaviors require context and access to education through the best and easiest channels, which seems to be appropriate for social media. Different educational approaches can be effective in reducing low back pain, disability and improving the health care workers life. The social media approach has been more successful than long-term face-to-face intervention and may be a better way to deliver training programs because of its ease of access and reduced operating costs
				factors, enabling factors and preventive behaviors of LBP in the control group (*p* > 0.05). Instruments: Visual Analogue Scale (VAS) for measuring LBP, for measuring pain-related disability, the Quebec Back Pain Disability Scale (QBPDS) was used. The social media approach to maintaining behavior for a long time (6 months) was more successful than the face-to-face approach	
Kozak et al. (2017) Germany [31]	To evaluate metrologically the effectiveness of a training program on the reduction of stressful trunk postures in geriatric nursing professions.	Pilot study	2-weeks before and 6-months after intervention Participants: 42	Outcomes: Measurement data were available from 23 participants at baseline and from 19 participants at a 6-month follow-up. After the intervention, the median proportion of time spent in sagittal inclinations exceeding 20° was significantly reduced, by 29% (*p* < 0.001), from 1772 to 1708 median trunk movements per shift. The proportion of very pronounced inclinations exceeding 60° was reduced by 60% (*p* = 0.002), from 288 to 135 inclinations per shift. A significant reduction in static inclinations was also detected (22%; *p* < 0.001), from 462 to 329 inclinations per shift (numbers of inclinations not in the table). The median time spent in sagittal inclinations exceeding 20° was reduced by 27 min per shift. esults of the video analyses at the second measure ment show that in total 217 basic care activities at the bedside were observed. As recommended by the seminar instructor, the bed was raised to hip height in 44.7% of all care situations. However, in 44.2% of situations, the bed was partially raised, and in 11.1%, the bed was not raised at all. In total, 52 care situations in the bathroom were observed. A stool was used in 67.3% of these situations to perform basic care in the sitting position; in 32.7% of the situations, the stool was not used by the nurses. Instruments: The CUELA measurement system and video analyses were used to evaluate this intervention.	This study showed a significant improvement in body postures after implementation of a training concept consisting of instruction on frequent body postures in nursing, physical exercises, instructions in practical ergonomic work at the bedside and in the bathroom, and reorganization of work environment.
Van Eerd et al. (2021) Canada [32]	To follow the implementation of a participatory organizational change intervention and assess the program implementation effects on important intermediate outcomes.	Mixed methods implementation	Baseline pre- intervention (Time 1—pre- implementation), 6 months (Time 2—mid-point of implementation), and 12 months (Time 3—end of implementation). Participants:197	Outcomes: there were 132 participants with 65 from the control sites and 67 from the intervention sites. Control group self-efficacy score did not increase over time whereas for the intervention group, self-efficacy scores increase over time. Three of the six measures (back (motion), shoulder/arm, wrist/hand) favored the intervention sites. Respondents concerns about safety and worker health most often concerned MSDs, stress levels and staff safety. They often referred to MSD hazards related to force and posture. They noted a culture of resistance to change, even though many felt the changes were likely to be beneficial Instruments: Three different methods: (1) self-administered questionnaires; (2) staff observations; (3) interviews and focus groups.	The increase in operational leadership confidence to address MSD hazards is important for programs implementation but also for program impact and sustainability. An important facilitator to implementing a participatory approach is early frontline staff involvement.
Garg et al. (2012) USA [33]	To evaluate long-term efficacy of an ergonomics program that included patient-handling devices in six long-term care facilities (LTC) and one chronic care hospital (CCH).	A pre- and postintervention design without a control group	During the intervention and postintervention. Participants: 504	Outcomes: “Compared with preintervention data, posintervention data showed significant improvements in injuries, lost workdays, modified-duty days, and workers ‘compensation costs associated with patient-handling activities. Decreased by 59.8%, lost workdays by 86.7%, modified-duty days by 78.8%, and workers ‘compensation cost by 90.6%. Instruments: For each nursing facility, mean values for patient-handling injuries, lost workdays, modified-duty days, and workers’ compensation costs were calculated per 100 nursing full-time equivalent employees (FTEs) per year for both pre- and postintervention. They rated each device for stresses to low back and shoulders on the Borg CR-10 Scale pacient comfort on a 7-point comfort scale and patient safety on a 7-point scale similar to Corlett and Bishop’s scale. They also rated the manual transfer method using gait belt for lofting and transferring patients from bed to wheelchair and draw sheet for repositioning in bed.	The study demonstrated that the implementation of patient-handling devices along with a comprehensive ergonomics program was effective in reducing injuries, lost workdays, modified-duty days, and workers compensation costs. They identified that for effective implementation of patient-handling devices, are no-manual-lifting policy, not readily available device, inadequate training of nursing, concerns for patient safety and longer transfer time with devices than with manual methods.
Evanoff et al. (2003) USA [34]	To evaluate the effectiveness of mechanical patient hoists at reducing musculoskeletal injuries following the deployment of such lifts in acute care hospitals and long-term care (LTC) facilities	A pre and post intervention study	Pre-intervention it was between January 1, 1996 and the date lifts were deployed-1997 or 1998, post-intervention was after the lifts were deployed until 31 December 2000. Participants: 412	Outcomes: “we observed 412 recordable musculoskeletal injuries during the study period, from a heath care worker population of 13,6 million productive hours (equivalent to 6835 full-time work years). Data combining the acute care and LTC units showed a RR of recordable injury of 0.82 (95% CI: 0.68–1.00), RR of lost day injury of 0.56 (0.41–0.78). And RR for lost day rate of 0.38, comparing the post intervention to the pre-intervention period. The injury rate decreased from 6.59 injuries annually per 100 FTE during the pre-intervention period to 5.70 injuries during the post-intervention period. Lost time rate also declined markedly in the post-intervention period. Instruments: Data on injuries and lost days were collected through OSHA 200 logs kept by each hospital. The interview asked how many times the worker had personally used a lift for transferring or weighing patients during the previous full-shift they worked, how many times they saw others use a lift for transferring or weighing patients on the previous shift, and why they did not use lifts more often.	The implementation of patient lifts can be effective in reducing occupational musculoskeletal injuries to nursing personnel in both LTC and acute care settings. In future work should focus on strategies to facilitate greater use of mechanical lifting device.
Li et al. (2004) USA [35]	To evaluate the effectiveness of mechanical patient lifts in reducing musculoskeletal symptoms, injuries, lost workday injuries, and workers’ compensation costs in workers at a community hospital	A pre and post intervention study	The pre-intervention period (July 1999 to January 2001), intervention period (August 2000 to January 2001), post-intervention period (February 2001 to March 2003). Participants:138	Outcomes: Of a total of 138 health care workers in the three intervention units, 61 (44%) completed the baseline symptom survey in June 2000. Compared with the pre-intervention period, post-intervention surveys showed statistically significant improvements in musculoskeletal comfort levels (*p*, 0.05) for all nine body parts surveyed. the nursing staff reported a statistically significant improvement in general health (0.2 out of a five point scale, *p* = 0.008). There was little change in the perceived intensity and difficulty of work, while statistically significant improvements were reported in job satisfaction, willingness to recommend their job to others, and in helpfulness of supervisor (all 0.4 out of a four point scale, *p*, 0.05. Instruments: At the training sessions, participants were given baseline ergonomic surveys (levels of musculoskeletal comfort of different body parts, the presence and severity of pain, and the levels of physical and mental exhaustion experienced ‘‘at the end of a typical workday’’). Other questions are information on several aspects of their work, such as the degree of support received from their supervisors, the amount of time available to complete tasks, and their level of job satisfaction.	Many nursing staff are reluctant to use mechanical lifts for patient handling tasks, the main reason reported was the lack of perceived need, followed by the lack of time and the lack of maneuvering space. They suggest that must be encouraged by management, and a policy of no manual lifting must be adopted.
Nguyen et al. (2022) Vietnam [36]	To evaluate the effectiveness of basic interventions (education, physical exercise) to prevent MSDs among district hospital nurses in Vietnam	A quasi-experimental before/after study	Pre- and postintervention evaluation. Participants: 290 nurses	Outcomes: “an intervention for 162 nurses and had a total of 128 nurses in the control group. Some general characteristics between the intervention group and the control group at baseline were quite similar, shown by *p* values in the Student´s *t*-test and the Chi2 test, both higher than 0.05. (…) Regarding impact on the prevalence of MSDs, there was a significant difference of the GEE´s test on the prevalence of MSDs in the last 7 days between the 2 groups before and after the intervention with the *p* value = 0.016. In more detail, the prevalence of MSDs in the last 7 days in the control group was 1.9 times higher than in the intervention group after the intervention. For the prevalence of MSDs in the last 12 months, the test did not provide significance by showing that the *p* value is equal to 0.059 (…) the intervention measures are probably effective in reducing the prevalence of MSDs at 4 anatomical sites: neck, shoulder/upper arm, wrists/hand, and lower back. Instruments: A Sociodemographic Questionnaire; The Standardized Nordic Questionnaire; The Short Form of the Quality-of-Life Enjoyment and Satisfaction Questionnaire (Q-LES-Q-SF); the Kessler Psychological Distress Scale.	The intervention measures are probably effective in reducing the prevalence of MSDs at these neck, shoulder/upper arm, wrists/hand, and lower back. One of the limitations of the educational intervention in this study is that it only provides of theoretical information and knowledge to nurses but does not monitor the application of these measurements by nurses in their actual work.

**Table 3 jpm-13-00185-t003:** Quality assessment for randomized controlled trials—ROB 2.0.

Study	D1	D2	D3	D4	D5	Overall
Yassi et al. (2001) [25]	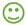	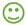	some concerns	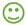	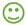	some concerns
Pourhaji et al. (2020) [30]	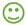	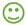	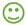	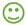	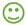	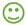

Legend: D1—Risk of bias arising from the randomization process; D2—Risk of bias due to deviations from the intended interventions (effect of assignment to intervention); D3—Missing outcome data; D4—4: Risk of bias in measurement of the outcome and D5—Risk of bias in selection of the reported result.

**Table 4 jpm-13-00185-t004:** Quality assessment of before–after studies with no control group.

Quality Assessment of before–after Studies with No Control Group					
	Nelson et al. (2006) [27]	Black et al. (2011) [28]	Garg et al. (2012) [33]	Evanoff et al. (2003) [34]	Li et al. (2004) [35]
Study question or objective clearly stated	Yes	Yes	Yes	Yes	Yes
Eligibility/selection criteria for the study population prespecified and clearly described	No	Yes	No	No	No
Participants in the study were representative of those who would be eligible for the intervention	No	No	Yes	Yes	Yes
Eligible participants were all enrolled	No	Yes	NR	NR	NR
The sample size was sufficiently large	Yes	Yes	Yes	Yes	Yes
The intervention was clearly described and delivered consistently	Yes	Yes	Yes	Yes	Yes
The outcome measures were pre-specified, clearly defined, valid, reliable, and assessed consistently	Yes	Yes	Yes	Yes	Yes
Outcomes were assessed blindly	Yes	Yes	No	Yes	No
The loss to follow-up after baseline was 20% or less	No	Yes		NR	
Were those lost to follow-up accounted for in the analysis?	Yes	Yes	NR	NR	NR
The statistical methods examine changes in outcome measures from before to after the intervention and statistical tests provided values for the pre-to-post changes	Yes	Yes	Yes	Yes	Yes
Outcome measures were taken multiple times before the intervention and multiple times after the intervention	No	No	No	No	No
If the intervention was conducted at a group level (e.g., a health unit, a community, Etc.), the statistical analysis took into account the use of individual-level data to determine effects at the group level	Yes	Yes	Yes	No	No

NR—Not reported.

**Table 5 jpm-13-00185-t005:** Quality assessment of observational and quasi-experimental study.

Quality Assessment of Papers Describing an Observational and Quasi-Experimental Study						
	Owen et al. (2002) [26]	Yang et al. (2021) [19]	Zadvinskis et al. (2009) [29]	Kozak et al. (2017) [31]	Van Eerd D et al. (2021) [32]	Nguyen et al. (2022) [36]
Was the study described as randomized, a randomized trial, a randomized clinical trial, or an RCT?	No	No	No	No	No	No
Was the method of randomization adequate (i.e., use of randomly generated assignment)?	No	No	No	No	No	Yes
Was the treatment allocation concealed (so that assignments could not be predicted)?	No	No	No	No	No	No
Were study participants and providers blinded to treatment group assignment?	NR	No	No	No	No	No
Were the people assessing the outcomes blinded to the participants’ group assignments?	NR	No	Yes	No	NR	No
Were the groups similar at baseline on important characteristics that could affect outcomes?	No	No	No	No	No	Yes
Was the overall drop-out rate from the study at endpoint 20% or lower than the number allocated to treatment?	No	No	NR	No	NR	NR
Was the differential drop-out rate (between treatment groups) at endpoint 15 percentage points or lower?	No	No	NR	No	NR	NR
Was there high adherence to the intervention protocols for each treatment group?	NA	NA	NA	NA	NA	NA
Were other interventions avoided or similar in the groups (e.g., similar background treatments)?	No	No	No	No	No	No
Were outcomes assessed using valid and reliable measures that were implemented consistently across all study participants?	Yes	Yes	Yes	Yes	Yes	Yes
Did the authors report that the sample size was sufficiently large to be able to detect a difference in the main outcome between groups with at least 80% power?	No	Yes	No	No	No	No
Were outcomes reported or subgroups analyzed prespecified (i.e., identified before analyses were conducted)?	No	No	Yes	No	No	Yes
Were all randomized participants analyzed in the group to which they were originally assigned, i.e., did they use an intention-to-treat analysis?	Yes	Yes	Yes	Yes	Yes	Yes

NA—Not applied; NR—Not reported.

**Table 6 jpm-13-00185-t006:** Type of interventions.

Study	Training Patient-Handling Devices	Ergonomics Education	Involving the Management Chain	Handling Protocol/Algorithms	Acquire Ergonomics Equipment’s	No-Manual Lifting	Others
Yassi et al. (2001) [22]	√	√					
Owen et al. (2002) [23]	√	√					
Nelson et al. (2006) [24]	√	√	√	√		√	
Black et al. (2011) [25]	√	√		√			
Yang et al. (2021) [16]	√	√		√	√		
Zadvinskis et al. (2009) [26]	√	√					
Pourhaji et al. (2020) [27]		√					MA; W/SN
Kozak et al. (2017) [28]	√	√			√		PE
Van Eerd D et al. (2021) [29]	√	√	√		N/M		
Garg et al. (2012) [30]	√	√			√		
Evanoff et al. (2003) [31]	√				√		
Li et al. (2004) [32]	√				√		
Nguyen et al. (2022) [33]	√	√		√			

√ - yes; N/M—Not Mentioned; MA—Mobile Application; W/SN—Website or Social Network; PE—Physical Exercise.

## Data Availability

Data are available only upon request to the authors.

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
