# Peer review of "The Effect of Interventions on Preventing Musculoskeletal Injuries Related to Nurses Work: Systematic Review"

_jpm, 2023, doi:10.3390/jpm13020185_

Round 1

Reviewer 1 Report (Previous Reviewer 2)

Thank you,

Mentioned changes and corrections have been made logically.

Author Response

 Dear Reviewer:

We appreciate your willingness to review our article and suggestions for improvement, if there are any more questions let we know.

Thanks

Reviewer 2 Report (New Reviewer)

Thanks to the editors for the opportunity to comment on the submitted article: “ The effect of Interventions on preventing musculoskeletal injuries related to nurses work: systematic review.“

I have a question/suggestion related to the abstract. In the abstract, I do not understand this formulation: "Method: The question was: What are the effects of musculo-skeletal disorders preventive interventions on nursing practice?" This is not a methodology. I recommend adding to the abstract the basic methods that were used in the elaboration of the topic. The conclusion of the abstract states: "This systematic review allows making recommendations for other studies." It would be appropriate to specify the results and give specific recommendations resulting from the review.

The authors work with appropriate literary sources (but can be extended e.g.https://doi.org/10.1111/hsc.13272 + https://doi.org/10.31482/mmsl.2021.007).

The submitted text can be considered as a professional review and I recommend it for publication after modifications.

Author Response

Dear Reviewer:

We appreciate your willingness to review our article and suggestions for improvement.

Review: 1 suggestions:

I have a question/suggestion related to the abstract. In the abstract, I do not understand this formulation: "Method: The question was: What are the effects of musculo-skeletal disorders preventive interventions on nursing practice?" This is not a methodology. I recommend adding to the abstract the basic methods that were used in the elaboration of the topic. The conclusion of the abstract states: "This systematic review allows making recommendations for other studies." It would be appropriate to specify the results and give specific recommendations resulting from the review.

Author´s:

Changes were made on the line 23, 24 and 25: "Method: This Systematic Review was guided by the research question ‘What are the effects of musculoskeletal disorders preventive interventions on nursing practice?"

Review: 2º suggestions

The authors work with appropriate literary sources (but can be extended e.g.https://doi.org/10.1111/hsc.13272 + https://doi.org/10.31482/mmsl.2021.007).

Author´s:

Thanks for your suggestion, we have read the articles, but they do not explore musculoskeletal disorders. However, two of the co-authors are researching people with long-term covid in Portugal and greatly appreciated the article - Selected social impacts and measures resulting from the Covid-19 epidemic in the Czech Republic on the specific example of the South Bohemian Region, and we will cite your sources, in the article they are preparing.

Thank you very much for your suggestions and help, if there are any more questions let me know.

This manuscript is a resubmission of an earlier submission. The following is a list of the peer review reports and author responses from that submission.

Round 1

Reviewer 1 Report

Dear Authors

This paper is about an interesting and significant issue for occupational health but after a careful review the decision is to make some revisions.

First, you can follow the PRISMA 2020 guideline for improving your reporting. I can not find the answers of the some questions below in your report.

INTRODUCTION

RATIONALE

-Describing the current state of knowledge and its uncertainties are not clear enough. You should improve your report it You should develop the reasons for the study by adding new literature sources.

-You should improve the introduction part by adding new literature resources to more clearly reveal the reasons for the study.

 -In explaining why it is important to do this review should be explained not only that the problem is important to nurses' health, but also where the lack of information is according to the available evidence.

- If other systematic reviews addressing the same (or a largely similar) question are available, explain why the current review was considered necessary.

-If the review examines the effects of interventions, also briefly describe how the intervention(s) examined might work.

- The study mentions the complexity of the interventions.  (If there is complexity in the intervention or context of its delivery (or both) (e.g. multi-component interventions, equity considerations)

RESEARCH QUESTİON 

Your research question  stated as “What are the effects of MDRW preventive interventions on nurses?”

-If the purpose is to evaluate the effects of interventions, you should use the Population, Intervention, Comparator, Outcome (PICO) framework or one of its variants, to state the comparisons that will be made. I can not find the Outcomes or Comparator in your research question

METHODS

ELIGIBILITY CRITERIA

-All study characteristics should be used to decide whether a study was eligible for inclusion in the review, that is, components described in the PICO framework or one of its variants, and other characteristics, such as eligible study design(s) and setting(s), and minimum duration of follow-up. Your eligibility criteria don't seem very clear and does not contain some of the information described above.

-You should specify eligibility criteria with regard to report characteristics, such as year of dissemination, language, and report status (e.g. whether reports, such as unpublished manuscripts and conference abstracts, were eligible for inclusion).

-Eligibility criteria of study does not include study period of articles or publication type/report status  such as original reports, thesis, unpublished report etc.

-Not clearly indicated if studies were ineligible because the outcomes of interest were not measured, or ineligible because the results for the outcome of interest were not reported.

In research report in results section only stated that “Regarding the year of publication, we can see that the oldest article was published in 141 2001 and that 38,5% of the articles were published in the last five years (one article in 142 2017, one article in 2 020, two in 2021, and one in 2022)”.

INFORMATION SOURCES

The problems in this study are:

-Not specified the date when each source (e.g. database, register, website, organisation) was last searched or consulted.

-We dont know the reference lists were examined or not, if yes, specify the types of references examined (e.g. references cited in study reports included in the systematic review, or references cited in systematic review reports on the same or similar topic).

SEARCH STRATEGY

- In addition to specifying bibliographic details of the reports to which citation searching was applied, the citation index or platform used (e.g. Web of Science), you should also give the date the citation searching was done.

DATA COLLECTION PROCESS

-If articles required translation into another language (you have different languages) to enable data collection, report how these articles were translated.

DATA ITEMS

- You should list and define the outcome domains and time frame of measurement for which data were sought.

- You should specify whether all results that were compatible with each outcome domain in each study were sought, and if not, what process was used to select results within eligible domains.

- You should consider specifying which outcome domains were considered the most important for interpreting the review’s conclusions and provide rationale for the labelling.

-Did you not consider the other variables? (e.g. participant and intervention characteristics). It could be important for interpretation of results of the review

RESULTS

RESULTS OF INDIVIDUAL STUDIES

-The summary of the studies included should focus only on study characteristics that help in interpreting the results (especially those that suggest the evidence addresses only a restricted part of the review question, or indirectly addresses the question).

-For all outcomes, irrespective of whether statistical synthesis was undertaken, present for each study an effect estimate and its precision (e.g. standard error or 95% confidence/credible interval).  You should mention about the effects estimate of the studies included in your report.  

PROTOCOL REGISTRATION

You reported the registration information “The protocol to guide the SR was prepared and agreed in December 2021 and is registered with Prospero under number 331581”. But when I check in Prospero, I see that the registration date is 31 May 2022 There is an incorrect information statement. And we understand that it is a retrospective recording, not a prospective one.

Author Response

INTRODUCTION

RATIONALE

Q: Describing the current state of knowledge and its uncertainties are not clear enough. You should improve your report it You should develop the reasons for the study by adding new literature sources.

-You should improve the introduction part by adding new literature resources to more clearly reveal the reasons for the study.

R: We made some changes, Page 2 line 61 to 77

Q: In explaining why it is important to do this review should be explained not only that the problem is important to nurses' health, but also where the lack of information is according to the available evidence.

R: We try to answer this question in page 2 line 93 to 95.

Q:  If other systematic reviews addressing the same (or a largely similar) question are available, explain why the current review was considered necessary.

-If the review examines the effects of interventions, also briefly describe how the intervention(s) examined might work.

R: We carried out this systematic review because we could not find another one that answered our research question.

Q: The study mentions the complexity of the interventions.  (If there is complexity in the intervention or context of its delivery (or both) (e.g. multi-component interventions, equity considerations)

R: The aim of the study was the identification of different interventions and analysis of their domains so that in the future we can design a multi-component intervention and assess its feasibility.

RESEARCH QUESTİON 

Q: Your research question stated as “What are the effects of MDRW preventive interventions on nurses?”

-If the purpose is to evaluate the effects of interventions, you should use the Population, Intervention, Comparator, Outcome (PICO) framework or one of its variants, to state the comparisons that will be made. I can not find the Outcomes or Comparator in your research question

R: On the PICO research strategy, it does not present directly, but in the option of observational studies PICO in the eligibility criteria for inclusion, the option for experimental and epidemiological studies with intervention, allowed the identification of studies with options for effectiveness and feasibility, and in strategy 3, the adoption by descriptors for preventative measures, intervention and assistance program, early intervention educational, allowed to obtain a comparison between different interventions.

METHODS

ELIGIBILITY CRITERIA

Q: All study characteristics should be used to decide whether a study was eligible for inclusion in the review, that is, components described in the PICO framework or one of its variants, and other characteristics, such as eligible study design(s) and setting(s), and minimum duration of follow-up. Your eligibility criteria don't seem very clear and does not contain some of the information described above.

R: We made some changes, Page 3 line 113 to 119

Q: You should specify eligibility criteria with regard to report characteristics, such as year of dissemination, language, and report status (e.g. whether reports, such as unpublished manuscripts and conference abstracts, were eligible for inclusion).

R: “There are no restrictions on country or year of research, but they were limited to studies published in Portuguese, English, and Spanish”. Page 3 line 116 to 119

Q: Eligibility criteria of study does not include study period of articles or publication type/report status such as original reports, thesis, unpublished report etc.

R: “Inclusion criteria were primary experimental and epidemiological study designs (RCT, non-RCT, quasi-experimental, cohort studies, case-control studies, and analytical cross-sectional studies)” page 3, line 113 to 115, no research was carried out in theses and publication reports

Q: Not clearly indicated if studies were ineligible because the outcomes of interest were not measured, or ineligible because the results for the outcome of interest were not reported

R: In section 2.2 we think we have answered this question

Q: In research report in results section only stated that “Regarding the year of publication, we can see that the oldest article was published in 141 2001 and that 38,5% of the articles were published in the last five years (one article in 142 2017, one article in 2 020, two in 2021, and one in 2022)”.

R: Regarding the year of publication, we can see that the oldest article was published in 2001 and that 38,5% of the articles were published in the last five years (one article in 2017, one article in 2020, two in 2021, and one in 2022). Page 6. line 181 to 183

INFORMATION SOURCES

The problems in this study are:

Q: Not specified the date when each source (e.g. database, register, website, organisation) was last searched or consulted.

R: We try to answer this question in page 3 line 141 to 144.

Q: We dont know the reference lists were examined or not, if yes, specify the types of references examined (e.g. references cited in study reports included in the systematic review, or references cited in systematic review reports on the same or similar topic).

R: The bibliographic references were consulted, but we didn´t find it necessary to extract any articles

SEARCH STRATEGY

Q: In addition to specifying bibliographic details of the reports to which citation searching was applied, the citation index or platform used (e.g. Web of Science), you should also give the date the citation searching was done.

R: It wasn´t carried out

DATA COLLECTION PROCESS

Q: If articles required translation into another language (you have different languages) to enable data collection, report how these articles were translated.

R: All selected articles were in English, and the research team is proficient in English.

DATA ITEMS

Q: You should list and define the outcome domains and time frame of measurement for which data were sought.

- You should specify whether all results that were compatible with each outcome domain in each study were sought, and if not, what process was used to select results within eligible domains.

- You should consider specifying which outcome domains were considered the most important for interpreting the review’s conclusions and provide rationale for the labelling.

-Did you not consider the other variables? (e.g. participant and intervention characteristics). It could be important for interpretation of results of the review

R: The changes in the method, and in the entire table for extracting the results allow us to respond to these suggestions. Changes were made. Page 7 to 12

RESULTS

RESULTS OF INDIVIDUAL STUDIES

Q: The summary of the studies included should focus only on study characteristics that help in interpreting the results (especially those that suggest the evidence addresses only a restricted part of the review question, or indirectly addresses the question).

R: We changed the results table and reformulate the content, so the studies characteristic’s are more clear and also the results that give answer to our aim.

Q: For all outcomes, irrespective of whether statistical synthesis was undertaken, present for each study an effect estimate and its precision (e.g. standard error or 95% confidence/credible interval).  You should mention about the effects estimate of the studies included in your report.

R: The confidence interval was 95%

PROTOCOL REGISTRATION

Q: You reported the registration information “The protocol to guide the SR was prepared and agreed in December 2021 and is registered with Prospero under number 331581”. But when I check in Prospero, I see that the registration date is 31 May 2022 There is an incorrect information statement. And we understand that it is a retrospective recording, not a prospective one.

R: The protocol to guide the SR was prepared and agreed in December 2021 and it was registered on Prospero with ID nº CRD42022331581 in May 2022. Submission Date 30 May 2022 page 3 line 108 to 109

Reviewer 2 Report

Dear Editorial Board,

Thank you for sending the manuscript entitled "The Effect of Interventions on preventing musculoskeletal injuries related to nurses’ work: systematic review" for review. This manuscript reports a systematic review addressing the topic of the interventions for MDRW prevention. In general, it is well explained and easy to read. I liked the topic of the study, it is quite interesting since the nurses need to know the prevention strategies and to be supported to use them.

I think it will be better if the author explains the following details:

Abstract:

1- Please provide a structured abstract including an introduction or background, aims, methods, results, and conclusion with a reasonable size and important details. For example, the abstract start with a paragraph and then a background. Or result part is a sentence. The study question and objective seem the same, so I think the study objective is enough.

2- Write the used different databases in the abstract.

Keywords:

3.  Please provide the related keywords according to MeSH.

Introduction

4. The first paragraph is better to omit because of a weak relation to MDRW. The authors should remark the kind of MDRW as low-back pain, restricting shoulder, knee, and wrist pain.

5.  Scientific writing is needed. Each paragraph needs at least 3-4 sentences. In the introduction, each sentence forms a paragraph.

6. You said about the different interventions (lines 48-52) so what is the gap in studies or lack of information about this. The identified research gap has to be mentioned. In my opinion, the introduction could focus more clearly on the rationale for this review study. There has been much systematic review on the MDRW among nurses. The authors should look at the literature for more information about the gap for another systematic review in this regard. As a starting point, they could consider

- Richardson et al (2018) Interventions to prevent and reduce the impact of musculoskeletal injuries among nurses: A systematic review. Int J Nurs Stud. 82: 58-67.  

- Dawson et al (2007) Interventions to prevent back pain and back injury in nurses: a systematic review. Occup Environ Med 64:642-50.

- Amick et al. Interventions in health-care settings to protect musculoskeletal health: a systematic review. Toronto: Institute for Work & Health. 2006

- Hignett S. Intervention strategies to reduce musculoskeletal injuries associated with handling patients: a systematic review. Occup Environ Med 2003;60:E6

-     Van Hoof et al (2018) The efficacy of interventions for low back pain in nurses: A systematic review. Int J Nurs Stud. 77:222-31.

-   Yassi et al (2013) Work-relatedness of low back pain in nursing personnel: a systematic review. Int J Occup Environ Health; 19:223-44.

-        And so on….

 7. Please explain PICO details. Your objective is to identify which interventions prevent MDRW in nurses. But your question is what the effects of MDRW preventive interventions are.

Methods

8. The outline of the methodology used is not adequate. Although the PRISMA checklist is cited, the method section does not follow it in its entirety. 

9.  Ordinarily, first, we search the Cochran to find if there is any duplication. Then, extend the search to the other databases.

10. (Line 109-110) Why did the writers focus on four databases? There is a lack of use of Google Scholar, Health Source Nursing, Emerald Insight, EMBASE, PsycInfo, Academic Search Complete, InformaWorld, and Oxford Journals for more comprehensive information.

11. Please transfer the Search Strategy (table 1) to a supplement file. In Table 1 you write the strategy used in Medline research instead of search. Writing the list of keywords in text is necessary according to PICO.

12.  Please explain how using MeSH could have helped in this regard.

13.  How did you control duplicates? By software like EndNote or manual?

Results

14.  Please write the references of each table below them.

15.  You wrote in the Table 1 that you find 1,106,849 article in strategy #1; 134 in # 2; and 1,294,797 in search strategy #3.

-  But why did you check the duplication for only 165 articles (line 128: A total of 48 duplicate articles (of the 165 articles submitted).

-     This huge excluded article needs more details.

-  You could explain the number of the records identified through database searching.

16.  Table 2. Type of outcomes and results of studies (p. 152). Please explain bellows:

- What is the difference between the results and outcome

-    It is better to write “attributes of studies”

-   In the first column, you wrote: Study/year/country. But have not mentioned the country name of the studies.

-       In the 3rd Column, you wrote Type of Study/Sample. But have not mentioned the Subjects (Did you mean that they all are nurses?)

- In the 5th Column, you wrote Outcomes and Instruments. But have not mentioned the outcomes for some studies (Did you mean that outcome measures?)

-  The name of the 6th column is better replaced with Results

17.  Please, explain the Quality assessment for randomized controlled trials - ROB 2,0. And D1 to D5 stand for what domains?

Discussion

18.  Please, in the first paragraph of the discussion part, write a sum of the study, for example, aims, number of studies, their quality, and results.

19.  The discussion must focus on the PICO and then aim. I was fairly confused by the discussion. However, the discussion part is good but needs rearrangement.

20.  One of the study limitations is the search of a few databases. I think it has to be mentioned in the limitations.

Conclusion

21.  Conclusion should be more relative to the study itself. How we can use these findings in the MDRW implementation includes educating nurses, changing practices to combine safety, peer education and support, partnership, empowerment, and cultural perceptions into everyday nursing practice.

Author Response

Abstract:

Q: Please provide a structured abstract including an introduction or background, aims, methods, results, and conclusion with a reasonable size and important details. For example, the abstract start with a paragraph and then a background. Or result part is a sentence. The study question and objective seem the same, so I think the study objective is enough.

Write the used different databases in the abstract.

R: The abstract was reorganized, and the databases used were added. Page 1 line 13 to 34

Keywords:

Q: Please provide the related keywords according to MeSH.

R: We introduced the terms MeSH page 1, line 35 to 36

Introduction

Q: The first paragraph is better to omit because of a weak relation to MDRW. The authors should remark the kind of MDRW as low-back pain, restricting shoulder, knee, and wrist pain.

R: We change the first paragraph of the introduction.

Q: Scientific writing is needed. Each paragraph needs at least 3-4 sentences. In the introduction, each sentence forms a paragraph.

R: We made some changes to give answer to this recommendation along the text.

Q: You said about the different interventions (lines 48-52) so what is the gap in studies or lack of information about this. The identified research gap has to be mentioned. In my opinion, the introduction could focus more clearly on the rationale for this review study. There has been much systematic review on the MDRW among nurses. The authors should look at the literature for more information about the gap for another systematic review in this regard. As a starting point, they could consider

R: We made some changes, Page 2 line 61 to 77 and page 2 and 3 line 93 to 99

Q: Please explain PICO details. Your objective is to identify which interventions prevent MDRW in nurses. But your question is what the effects of MDRW preventive interventions are.

R: We made some changes, page 2 line 96 to 99. On the PICO research strategy, it does not present directly the Outcomes, but in the option of observational studies PICO in the eligibility criteria for inclusion, the option for experimental and epidemiological studies with intervention, allowed the identification of studies with options for effectiveness and feasibility, and in strategy 3, the adoption by descriptors for preventative measures, intervention and assistance program, early intervention educational, allowed to obtain a comparison between different interventions.

Methods

Q: The outline of the methodology used is not adequate. Although the PRISMA checklist is cited, the method section does not follow it in its entirety. 

Ordinarily, first, we search the Cochran to find if there is any duplication. Then, extend the search to the other databases.

(Line 109-110) Why did the writers focus on four databases? There is a lack of use of Google Scholar, Health Source Nursing, Emerald Insight, EMBASE, PsycInfo, Academic Search Complete, InformaWorld, and Oxford Journals for more comprehensive information.

R: We use the databases available at our institution. For example, EMBASE, which would be an important base in the health field, is not available. Regarding academic google, it was not an option because we consider that given the type of studies we wanted and peer-reviewed, we were able to obtain them in the databases for scientific articles.

Q: Please transfer the Search Strategy (table 1) to a supplement file. In Table 1 you write the strategy used in Medline research instead of search. Writing the list of keywords in text is necessary according to PICO.

R: In the journal, we did research on other articles and they usually present the research strategy made in at least one database in the body of the method, so we kept.

Q: Please explain how using MeSH could have helped in this regard.

R: The use of MeSH terms helps because of the indexing of articles in the medline, since many journals recommend that the article descriptors be those of the MeSH

Q: How did you control duplicates? By software like EndNote or manual?

R: We made some changes, page 5, line 149 to 154

Results

Q: Please write the references of each table below them.

You wrote in the Table 1 that you find 1,106,849 article in strategy #1; 134 in # 2; and 1,294,797 in search strategy #3.

R: This is the search strategy for MEDLINE, via PUBMED, Then we search in other databases and obtain some articles in duplicate.

Q: But why did you check the duplication for only 165 articles (line 128: A total of 48 duplicate articles (of the 165 articles submitted).  This huge excluded article needs more details.

R: There are articles that we obtained in different databases, because they are indexed in different platforms.

Q: You could explain the number of the records identified through database searching.

R: We try to answer this question on page 5, line 150 to 155

Q: Table 2. Type of outcomes and results of studies (p. 152). Please explain bellows:

- What is the difference between the results and outcome

R: We change the table and present the outcomes (results of the studies) and the conclusions

Q: It is better to write “attributes of studies”. In the first column, you wrote: Study/year/country. But have not mentioned the country name of the studies.

R: We made some changes, because we add this information, and the changes give the answer to this recommendation.

Q: In the 3rd Column, you wrote Type of Study/Sample. But have not mentioned the Subjects (Did you mean that they all are nurses?)

R: We add this information, 

Q: In the 5th Column, you wrote Outcomes and Instruments. But have not mentioned the outcomes for some studies (Did you mean that outcome measures?)

R: We made some changes.

Q: The name of the 6th column is better replaced with Results

R: We made some changes in table 2, page 7 to 12

Q: Please, explain the Quality assessment for randomized controlled trials - ROB 2,0. And D1 to D5 stand for what domains?

 R: We add a Legend after the table: D1 - Risk of bias arising from the randomization process; D2 - Risk of bias due to deviations from the intended interventions (effect of assignment to intervention); D3 - Missing outcome data; D4 - 4: Risk of bias in measurement of the outcome and D5- Risk of bias in selection of the reported result.

Discussion

Q: Please, in the first paragraph of the discussion part, write a sum of the study, for example, aims, number of studies, their quality, and results.

R: We add the first paragraph with this information.

Q: The discussion must focus on the PICO and then aim. I was fairly confused by the discussion. However, the discussion part is good but needs rearrangement.

R: We focused the discussion on the interventions included in the different studies.

Q: One of the study limitations is the search of a few databases. I think it has to be mentioned in the limitations.

R: We add it.

Conclusion

Q: Conclusion should be more relative to the study itself. How we can use these findings in the MDRW implementation includes educating nurses, changing practices to combine safety, peer education and support, partnership, empowerment, and cultural perceptions into everyday nursing practice.

R: We change it, and we add this information in the conclusion.

Round 2

Reviewer 1 Report

Dear Authors

There is some unanswered questions and also some important problems

1-If the review examines the effects of interventions, also briefly describe how the intervention(s) examined might work (UNRESPONDED!!!)

2-“What are the effects of MDRW preventive interventions on nurses?” (UNSATISFYING)

You should mention about outcomes!!!

İf comparator is a general term (it means you can accept all comparator-active or passive) no need to mention about.

3-Q: Not clearly indicated if studies were ineligible because the outcomes of interest were not measured, or ineligible because the results for the outcome of interest were not reported

R: In section 2.2 we think we have answered this question (YOU HAVE TO SPECIFY THE LINE!!!)

4-Q:You should mention about the effects estimate of the studies included in your report.

-R: The confidence interval was 95% (YOU SHOULD ADD EFFECT ESTIMATE AND CI OF STUDIES INCLUDED IN YOUR STUDY!!!)

5-Q: You reported the registration information “The protocol to guide the SR was prepared and agreed in December 2021 and is registered with Prospero under number 331581”. But when I check in Prospero, I see that the registration date is 31 May 2022 There is an incorrect information statement. And we understand that it is a retrospective recording, not a prospective one.

R: The protocol to guide the SR was prepared and agreed in December 2021 and it was registered on Prospero with ID nº CRD42022331581 in May 2022. Submission Date 30 May 2022 page 3 line 108 to 109

-YOU SHOULD ALWAYS GIVE THE DATE OF SUBMISSION NOT PREPARATION TIME ?????

-THIS IS STILL A MISINFORMATION ABOUT YOUR STUDY REPORTING. AND IN MY OPINION IT IS AN ETHICAL PROBLEM ABOUT YOUR STUDY

-ALSO, RETROSPECTIVE REGISTRATION IS A PROBLEM FOR SELECTIVE REPORTING???

Author Response

Dear Reviewer.

First, we want to thank you for your orientation, comments, and suggestions, about this systematic review, because Increases scientific rigor and guides us towards ethical issues to avoid in future SRs.

Respond to reviewer question:

  1. If the review examines the effects of interventions, also briefly describe how the intervention(s) examined might work (UNRESPONDED!!!)

R: We conclude based on the effectiveness of the interventions that: “multifactor training (transfer, lifting, and repositioning), and the multiple interventions (education and training; zero lift policy; provision of assistive devices for patient support and care; individual measure, etc), are emerging as the most effective instruments in the prevention of MDRW” (page 17, lines 306 to 310)

“A peer leader program is much more effective than traditional educational approaches and facilitates the implementation of the program as well as sustaining over time, especially in small hospitals.” (page 17, lines 314 to 316)

  1. “What are the effects of MDRW preventive interventions on nurses?”

R: We identify the effects of MDRW preventive interventions on nurse in:

“ These multidimensional intervention programs, reduce the self-reported performance of “unsafe” working environment, decrease time-loss /injury days, modify duty days, increase job satisfaction, and decrease workers compensation costs.” (page 17, lines 311 to 313)

“The procede-proceed model has a significant effect on behaviors as a factor that increases the quality of lifestyles of low back pain (LBP). Theoretical education was effective in improving knowledge, attitude, and self-efficacy, reforcing and enabling factors, and behavior immediately after 6-12 months of intervention. Other study conclude that the effective goal of reducing MDRW, is the combining of theorical edu-cation with ergonomics practice. The social media approach to maintaining be-havior for long time (6 months) was more successful than face-to-face approach.” (page 17, lines 317 to 323).

  1. Not clearly indicated if studies were ineligible because the outcomes of interest were not measured, or ineligible because the results for the outcome of interest were not reported

R: Sorry for not indicating the page and line. The answer to this question is on page 3, lines 118 to 126.

  1. You should mention about the effects estimate of the studies included in your report.

R: Sorry. The answer to this question is on page 5, lines 165 to 166.

  1. You reported the registration information “The protocol to guide the SR was prepared and agreed in December 2021 and is registered with Prospero under number 331581”. But when I check in Prospero, I see that the registration date is 31 May 2022 There is an incorrect information statement. And we understand that it is a retrospective recording, not a prospective one

R: Regarding the registration of the review, you are right about the delay in the registration (Prospero – CRD42022331581) and the time of submission of the RS to MDPI.

We did the protocol for RS and filled all the submission requirements, but there was a forgetfulness in the validation of the submission. We started the review, and when we detect the mistake, we had one month to submit the RS to MDPI. We immediately proceed to the submission and indicate that we were already developing the review.

We obtained the consent 30 days later (31/05/2022), and our deadline for MDPI was 30/5/2022 (the day that we submitted the RS).

We know that it is not the rate way to do, but this was the truth, and it was a lesson for life. (page 3, lines 110 to 111).

Reviewer 2 Report

Thank you for applying the comments,

I am confused about your aim yet, as you wrote the aim of this review was to identify the different interventions adopted in the prevention of MDRW in nurses, to compare the effectiveness of the interventions to prevent MDRW in nurses, and provide ….

However in the discussion and conclusion parts, you only identified the different interventions such as (lines: 312- 313) training patient handling devices; ergonomics education; involving the management chain; handling protocol/algorithms; acquiring ergonomics equipment, and no-manual lifting.

There is no comparison between the different interventions' effectiveness.

Author Response

Hello Dear Review.

To respond to your question about the aim of the SR, we added on the abstract (page 1, lines 31 to 32); on discussion (page 17, lines 306 to 323); and the conclusion (page 18, lines 338 to 340).

We want to thank for your orientation, comments and suggestions, about this systematic review, because increased the rigor of the SR.
